# Development of a multiplex real-time PCR assay for the simultaneous detection of four bacterial pathogens causing pneumonia

Ho Jae Lim[1,2☯], Eun-Rim Kang[1☯], Min Young Park[1], Bo Kyung Kim[3], Min Jin Kim[3], Sunkyung Jung[1], Kyoung Ho Roh[4], Nackmoon Sung[5], Jae-Hyun Yang[6], Min-Woo Lee[7], Sun-Hwa Lee[1,3], Yong-Jin Yang[1]*

1 Department of Molecular Diagnostics, Seegene Medical Foundation, Seoul, Republic of Korea,
2 Department of Integrative Biological Sciences, Chosun University, Gwangju, Republic of Korea,
3 Department of Laboratory Medicine, Seegene Medical Foundation, Seoul, Republic of Korea,
4 Department of Laboratory Medicine, National Health Insurance Service Ilsan Hospital, Goyang, Gyeonggi, Republic of Korea, 5 Clinical Research Institute, Seegene Medical Foundation, Seoul, Republic of Korea,
6 Paul F. Glenn Center for Biology of Aging Research, Department of Genetics, Blavatnik Institute, Harvard Medical School, Boston, MA, United States of America, 7 Soonchunhyang Institute of Medi-bio Science (SIMS) and Department of Integrated Biomedical Science, Soonchunhyang University, Cheonan-si, Republic of Korea

☯ These authors contributed equally to this work.
* yjyang@mf.seegene.com

**Data Availability Statement:** All relevant data are within the paper and its Supporting Information files.

## Abstract

Classification of clinical symptoms and diagnostic microbiology are essential to effectively employ antimicrobial therapy for lower respiratory tract infections (LRTIs) in a timely manner. Empirical antibiotic treatment without microbial identification hinders the selective use of narrow-spectrum antibiotics and effective patient treatment. Thus, the development of rapid and accurate diagnostic procedures that can be readily adopted by the clinic is necessary to minimize non-essential or excessive use of antibiotics and accelerate patient recovery from LRTI-induced damage. We developed and validated a multiplex real-time polymerase chain reaction (mRT-PCR) assay with good analytical performance and high specificity to simultaneously detect four bacterial pathogens causing pneumonia: *Klebsiella pneumoniae*, *Pseudomonas aeruginosa*, *Staphylococcus aureus*, and *Moraxella catarrhalis*. The analytical performance of mRT-PCR against target pathogens was evaluated by the limit of detection (LOD), specificity, and repeatability. Two hundred and ten clinical specimens from pneumonia patients were processed using an automatic nucleic acid extraction system for the "respiratory bacteria four" (RB4) mRT-PCR assay, and the results were directly compared to references from bacterial culture and/or Sanger sequencing. The RB4 mRT-PCR assay detected all target pathogens from sputum specimens with a coefficient of variation ranging from 0.29 to 1.71 and conservative LOD of DNA corresponding to $5 \times 10^2$ copies/reaction. The concordance of the assay with reference-positive specimens was 100%, and additional bacterial infections were detected from reference-negative specimens. Overall, the RB4 mRT-PCR assay showed a more rapid turnaround time and higher performance that those of reference assays. The RB4 mRT-PCR assay is a high-throughput and reliable tool that assists decision-making assessment and outperforms other standard

**Funding:** The authors received no specific funding for this work.

**Competing interests:** The authors have declared that no competing interests exist.

methods. This tool supports patient management by considerably reducing the inappropriate use of antibiotics.

## Introduction

According to the World Health Organization (WHO), lower respiratory tract infections (LRTIs) were the main cause of morbidity and mortality leading to 3 million deaths worldwide in 2016 [1]. LRTIs, caused by various bacteria and viruses, are associated with different clinical symptoms and etiologies affected by age, sex, and season [2]. Owing to complicated clinical symptoms, efforts to reduce the global burden of LRTIs using preventive and treatment strategies require timely identification of pathogens [3].

LRTIs fall into two categories based on their origin: community-acquired pneumonia (CAP) and hospital-acquired pneumonia (HAP) [4]. In the Asia-Pacific region, 11 species of bacteria and eight species of viruses causing CAP are examined by serology, culture, immunofluorescence assay, or polymerase chain reaction (PCR) [5]. Notably, in LRTI patients with severe CAP, *Staphylococcus aureus*, *Klebsiella pneumoniae*, and *Pseudomonas aeruginosa* are the most common species, accounting for 28.6%, 28.6%, and 17.9%, respectively [6], of infections and are identified by diagnostic culture methods. In addition, despite low incidence, *Moraxella catarrhalis* has been recently recognized as an important emerging pathogen because it shows prevalent resistance to some antibiotics through beta-lactamase production [7].

Rapid identification of pathogens and determination of susceptibility profiles are critical for prompting appropriate therapy [8]. Nevertheless, the current standard diagnosis to detect respiratory bacteria is a culture-based procedure, which typically takes 24–72 h to process. Culture-based diagnosis of LRTI specimens has low sensitivity and negative predictive value (NPV); only approximately 30% of CAP patients receive correct microbiological identification [9]. Therefore, a combination of different tests, such as culture, antigen, and serology detection of pathogens, is commonly requested [10]. Thus, as the demand of rapid and highly precise methods for microbial identification increases, PCR-based assays would be an appropriate system to allow patient specimens to be examined on a large scale easily within quarter of a day [11, 12].

Current molecular diagnostic panels lack coverage of diverse respiratory pathogens. In the last few decades, conventional PCR assays have been performed for rapid identification of a wide range of pathogens; however, it requires the tube to be opened for post-detection analysis, leading to putative cross-contamination of the amplicon and consequent false-positive results [13]. Recently, the TaqMan method (Applied Biosystems, Inc., CA, USA) has been developed in the multiplex real-time PCR (mRT-PCR) platform, enabling quantitative applications [14]. The mRT-PCR assay is highly sensitive, specific, rapid, and less prone to false-positives [13].

Here, we developed a high-throughput mRT-PCR assay for the rapid and reliable identification of key respiratory bacterial pathogens causing pneumonia: *S. aureus*, *M. catarrhalis*, *K. pneumoniae*, and *P. aeruginosa*. Direct comparison to bacterial culture and/or a sequencing method (reference assays) in LRTI specimens was used to validate the effectiveness of our high-throughput respiratory bacteria four (RB4) mRT-PCR assay.

## Materials and methods

### Primer/PROBE design

Candidate assay targets for the four key bacteria were selected based on publications. National Center for Biotechnology Information basic local alignment search tool (NCBI-BLAST) for

**Table 1. Oligonucleotide primers and probes for the real time PCR to detect target pathogens and an internal control.**

| Pathogen | Target gene | Type | Sequences (5'-3') | $T_m$ (°C) | Product size (bp) | Reference |
|---|---|---|---|---|---|---|
| *K. pneumoniae* | *yphG* | F. primer | GAGTTAGGGAAACGAACATTGTGIIIIIGGCAGTGCTC | 58 | 137 | [16] |
| | | R. primer | TCTCTATCGGACAGACGTCGGIIIIIAAAGAGGTTC | 59 | | |
| | | probe | [a]FAM-TTCATTGGCATCATCACTTAGCGAC | 65.1 | | |
| *P. aeruginosa* | *regA* | F. primer | GCTTCATCGACAGCATCGGIIIIIACTGACGCCA | 58.1 | 111 | [17] |
| | | R. primer | CGGCTTTTTCTCTGGTCTGTGIIIIIGTTCCGCTGT | 59.1 | | |
| | | probe | [b]HEX-AACACAAACGCACTCGGAAAAATCG | 68.5 | | |
| *S. aureus* | *nuc* | F. primer | GATGGCTATCAGTAATGTTTCGAIIIIIICAATACRCAA | 56.3 | 213 | [18] |
| | | R. primer | GTCGCAGGTTCTTTATGTAATTTTIIIIITGAAGTTGCA | 57.4 | | |
| | | probe | [c]Texas red-CAAGTCTAAGTAGCTCAGCARATGCATCA | 67.1 | | |
| *M. catarrhalis* | *copB* | F. primer | ATTCGTGGCATGGGTCATAAT | 58.5 | 182 | [9] |
| | | R. primer | GTAACAATCGCACCRTTGGTT | 56.8~59.3 | | |
| | | probe | [d]Cy5.5-CACCAAGGTCGCTTTATGCTAGACCC | 68.7 | | |
| Internal control | *HBB* | F. primer | GGCATAAAAGTCAGGGCAGAIIIIIICTATTGCT | 56.9 | 158 | - |
| | | R. primer | CCAACTTCATCCACGTTCACCIIIIIICCACAGGG | 59.0 | | |
| | | probe | [e]Cy5-CCTGAGGAGAAGTCTGCCGTTACTGC | 68.8 | | |

Probes were labeled with FAM, HEX, Texas red, Cy5.5, and Cy5 and detected at 520, 556, 616, 694, and 669 nm, respectively. All the probes were had BHQ as a quencher at 3' end.

Abbreviations: $T_m$, melting temperature; F. primer, forward primer; R. primer, reverse primer; probe, fluorescently labeled primer; R: A or G.

nucleotide sequences was used to compare candidate oligonucleotides to targets in GenBank, including large-scale sequence databases. Assays were redesigned or modified using GeneRunner version 6.0 and NCBI-BLAST to optimize for multiplex performance. The specificity of optimized oligonucleotide sequences was analyzed *in silico* by the multAlin interface. Sequences were also screened by alignments within sequence databases for species selectivity. Considering these assessments, four targets were selected for the pathogen assay and/or designed for a dual priming oligonucleotide-based multiplex PCR assay for specificity [15]. The sequences of primers and probes are described in Table 1.

## Control isolates

The positive control for the four bacterial strains used in confirmation assays were as follows: *S. aureus* for American Type Culture Collection (ATCC) 29213, *M. catarrhalis* for Korean Culture Center of Microorganisms (KCCM) 42706, *K. pneumoniae* for ATCC 13883, and *P. aeruginosa* for ATCC 27853. Plasmids containing target genes were generated by cloning amplicons with the pLUG-prime TA-cloning vector system (iNtRON Biotechnology, Inc., Seongnam, Korea). Plasmid DNA was diluted in DNA-spin (iNtRON) to 1 copy/reaction via serial dilution for PCR optimization and quantification standards. *S. auricularis* and *S. haemolyticus* were the predominantly isolated respiratory specimens in Seegene Medical Foundation (SGMF) for use as a negative control. Most of these controls were commercially supplied as DNA extracts from ATCC, Korean Collection for Type Cultures (KCTC), KCCM, and Korea Bank for Pathogenic Viruses (KBPV).

## Nucleic acid extraction

All patient samples were simultaneously processed using an automated nucleic acid extraction system, the MagNA PURE 96 (Roche, Inc., Basel, Switzerland), according to the manufacturer's instructions. The specimens were pre-treated with 1 mL of phosphate-buffered saline

solution (Biosesang, Co., Seongnam, Korea) and vigorously vortexed for 10 s. Each sample was divided into 200 µL for nucleic acid extraction. Nucleic acid samples were eluted with 100 µL elution buffer and stored at −20˚C until use.

## Multiplex real-time PCR

PCR procedures were carried out using the CFX96 instrument (Bio-Rad Laboratories, Inc., Irvine, CA, USA). RB4 mRT-PCR assays were performed in a total reaction volume of 20 µL, comprising µL oligonucleotide mixture, 5 µL 4X PCR enzyme, and 5 µL of nucleic acid extract. PCR was conducted with the following parameters: 95˚C for 15 min in the first step, followed by 38 cycles of 95˚C for 10 s for denaturation and 60˚C for 60 s for annealing/extension. Test runs were validated when positive controls for each amplification target were positive and negative controls (no template) were negative. The RB4 mRT-PCR assay is designed to detect *K. pneumoniae* on FAM, *P. aeruginosa* on HEX, *S. aureus* on Texas red, and *M. catarrhalis* on the Cy5.5 channel. To prevent false-negative results, the human hemoglobin subunit beta (*HBB*) was simultaneously amplified and was the detected housekeeping gene used as an endogenous internal control.

## Analytical performance

The LOD was obtained from the positive control assessments for *K. pneumoniae*, *P. aeruginosa*, *S. aureus*, and *M. catarrhalis*. Each control was serially diluted into $10^3$, $5 \times 10^2$, $10^2$, $5 \times 10^1$, $10^1$, and 1 copies/reaction. LOD tests were performed 50 times for these concentrations. Likewise, repeatability tests were determined 20 times for intra-assay coefficients of variation (CV): high for $10^3$ copies/reaction, medium for $5 \times 10^2$ copies/reaction, low for $10^2$ copies/reaction, and very low for $5 \times 10^1$ copies/reaction. Fifty strains of bacteria and 19 strains of viruses were selected for reactivity tests.

## Clinical LRT specimens

A total of 210 anonymized residual sputum specimens from patients presenting symptoms of pneumonia were obtained and preserved for routine procedures between June and August 2018. All specimens were classified into two groups (positive or negative) with reference assays. Samples were confirmed as positive if more than one result from reference assays were positive.

## Microbial identification

Well-mixed sputum specimens were cultured on blood and MacConkey agar plates. The plates were incubated for 24 h at 37˚C. The colonies from culture plates were deposited on an assay plate, and 1 µL 70% formic acid and 1 µL matrix solution were added. Thereafter, the plate was analyzed using a Bruker Biotyper MALDI-TOF (Bruker, Bremen, Germany).

## Sanger sequencing

In the event of a discrepancy between culture assays and PCR assays, Sanger sequencing was performed as an additional confirmatory test in a different institute (Cosmogenetech, Co., Seoul, Korea). The results matched, and over 95% were regarded as references. Sequencing data were analyzed using NCBI-BLAST.

## Statistical analysis

All statistical analyses were performed using SPSS version 26.0 (IBM, Co., NY, USA) for Windows. The analytical Cohen's kappa defined statistical significance only if $P$-values were $\leq 0.05$. The CVs were determined for mRT-PCR platforms using measurements obtained 20 times from double runs and presented as means and standard deviations. The LOD of mRT-PCR assays, the concentration of the sample detected as positive with 95% confidence, was estimated to fit the probit regression model. For the diagnostic test, sensitivity, specificity, positive predictive values (PPVs), and NPVs were used to compare each RB4 mRT-PCR assay to the reference result and subsequently estimate the diagnostic accuracy of the pathogens.

## Ethics statement

Ethical aspects for this study were reviewed and approved by the Seegene Medical Foundation Institutional Review Board (approval number, SMF-IRB-2021-001), provided that after conducting the original test, the remaining anonymous sputum specimens were used. All data were fully anonymized administrative data without patient identifiers, and patient consent was waived by the institutional review board.

# Results

## Design of the RB4 mRT-PCR assay

The four pathogens selected for the RB4 mRT-PCR assay, *K. pneumoniae*, *P. aeruginosa*, *S. aureus*, and *M. catarrhalis*, are species with clinical significance and were detected at 1.3–24.7%, 0–8%, 0.4–10.4%, 0.3–15%, respectively, from pneumonia patients [5]. *yphG* and *regA* were selected for the detection of *K. pneumoniae* and *P. aeruginosa*, respectively, because they have been evaluated with great sensitivity in the literature [16, 17]. A thermostable nuclease gene, *nuc*, was selected to detect *S. aureus*, owing to its greater specificity [18]. A highly conserved region of *copB* was selected for *M. catarrhalis* detection [9], and *HBB* was used as an internal control (Table 1). All target genes were efficiently amplified, without inhibiting one another, in the PCR (Fig 1).

## Performance of analytical specificity

To confirm the specificity of the RB4 mRT-PCR assay, other bacterial and viral strains were tested (Table 2). The coverage of the four strains was 100% for each assay target, and no cross-reactivity was observed, indicating great specificity to all targets. The specificity against the 62 negative controls was 100%.

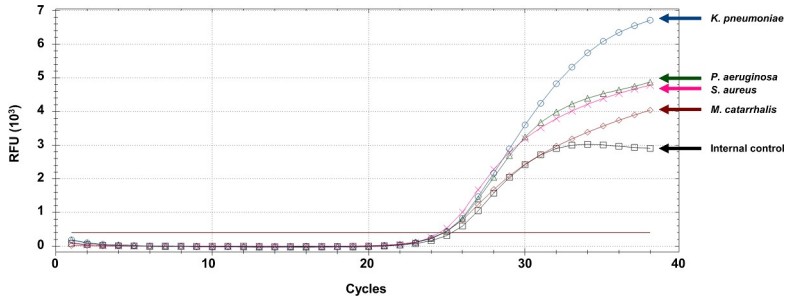

**Fig 1. RB4 mRT-PCR assay performed using positive controls corresponding to *Klebsiella pneumoniae*, *Pseudomonas aeruginosa*, *Staphylococcus aureus*, *Moraxella catarrhalis*, and the internal control, respectively.** The data were analyzed from cycle 1 to 38.

**Table 2. Sixty-nine isolates of global origin for analytical specificity of target pathogens.**

| Group | organism | Source | FAM | HEX | Cal Red 610 | Quasar 705 |
|---|---|---|---|---|---|---|
| Bacteria | *Klebsiella pneumoniae* | ATCC 13883 | + | - | - | - |
| | *Klebsiella pneumoniae* | KCCM 42750 | + | - | - | - |
| | *Pseudomonas aeruginosa* | ATCC 27853 | - | + | - | - |
| | *Pseudomonas aeruginosa* | KCCM 11266 | - | + | - | - |
| | *Staphylococcus aureus* | ATCC 29213 | - | - | + | - |
| | *Staphylococcus aureus* | KCCM 32395 | - | - | + | - |
| | *Moraxella catarrhalis* | KCCM 42706 | - | - | - | + |
| | *Corynebacterium glucuronolyticum* | ATCC 51860 | - | - | - | - |
| | *Enterobacter aerogenes* | ATCC 13048 | - | - | - | - |
| | *Enterobacter aerogenes* | KCCM 12177 | - | - | - | - |
| | *Enterobacter cloacae* | ATCC 13047 | - | - | - | - |
| | *Enterococcus faecalis* | KCTC 5290 | - | - | - | - |
| | *Enterococcus faecalis* | KCCM 12117 | - | - | - | - |
| | *Enterococcus faecium* | KCTC 13225 | - | - | - | - |
| | *Haemophilus influenzae* | ATCC 9007 | - | - | - | - |
| | *Klebsiella ornithinolytica* | KCCM 41044 | - | - | - | - |
| | *Klebsiella oxytoca* | ATCC 13182 | - | - | - | - |
| | *Klebsiella planticola* | KCCM 11649 | - | - | - | - |
| | *Klebsiella quasipneumoniae sub* | ATCC 700603 | - | - | - | - |
| | *Lactobasillus crispatus* | ATCC 33820 | - | - | - | - |
| | *Lactobasillus gasseri* | ATCC 33323 | - | - | - | - |
| | *Lactobasillus jensenii* | ATCC 25258 | - | - | - | - |
| | *Moraxella atlantae* | KCCM 11242 | - | - | - | - |
| | *Moraxella osloensis* | KCTC 52865 | - | - | - | - |
| | *Plasmodium falciparum* | ATCC 30932 | - | - | - | - |
| | *Proteus mirabilis* | ATCC 29906 | - | - | - | - |
| | *Pseudomonas alcaligenes* | KCTC 12029 | - | - | - | - |
| | *Pseudomonas basaltis* | KCTC 22136 | - | - | - | - |
| | *Pseudomonas fluorescens* | KCTC 1767 | - | - | - | - |
| | *Pseudomonas putida* | KCTC 1642 | - | - | - | - |
| | *Pseudomonas taetrolens* | KCTC 12501 | - | - | - | - |
| | *Staphylococcus auricularis* | ATCC 33753 | - | - | - | - |
| | *Staphylococcus caprae* | KCTC 3583 | - | - | - | - |
| | *Staphylococcus epidermidis* | Clinical isolate | - | - | - | - |
| | *Staphylococcus gallinarum* | KCTC 3585 | - | - | - | - |
| | *Staphylococcus haemolyticus* | Clinical isolate | - | - | - | - |
| | *Staphylococcus pasteuri* | KCTC 13167 | - | - | - | - |
| | *Staphylococcus Saprophyticus* | KCTC 3345 | - | - | - | - |
| | *Streptococcus agalactiae* | ATCC 29213 | - | - | - | - |
| | *Streptococcus downei* | KCTC 5808 | - | - | - | - |
| | *Streptococcus gordonii* | KCTC 3671 | - | - | - | - |
| | *Streptococcus mitis* | ATCC 29213 | - | - | - | - |
| | *Streptococcus mutans* | KCTC 5365 | - | - | - | - |
| | *Streptococcus oralis* | KCTC 5672 | - | - | - | - |
| | *Streptococcus parasanguinis* | ATCC15912 | - | - | - | - |
| | *Streptococcus pneumoniae* | ATCC 49619 | - | - | - | - |
| | *Streptococcus pyogenes* | KCTC 19615 | - | - | - | - |
| | *Streptococcus sanguinis* | KCTC 3284 | - | - | - | - |
| | *Streptococcus pneumoniae* | KCCM 40410 | - | - | - | - |
| | *Haemophilus influenzae* | KCCM 42099 | - | - | - | - |

(*Continued*)

**Table 2.** (Continued)

| Group | organism | Source | FAM | HEX | Cal Red 610 | Quasar 705 |
|---|---|---|---|---|---|---|
| Virus | BK virus | ATCC VR837 | - | - | - | - |
| | Coxasackie virus B3 | ATCC VR30 | - | - | - | - |
| | Dengue virus type 2 | ATCC VR1584 | - | - | - | - |
| | Dengue virus type 4 | ATCC VR1254CAF | - | - | - | - |
| | Herpes virus type 2 | ACTT VR734 | - | - | - | - |
| | Human Adenovirus type 18 | ATCC VR1095 | - | - | - | - |
| | Human Adenovirus type 40 | ATCC VR931 | - | - | - | - |
| | Human Adenovirus type 8 | ATCC VR1368 | - | - | - | - |
| | Human coronavirus OC43 | ATCC VR1558 | - | - | - | - |
| | Human Parainfluenzavirus | ATCC VR1380 | - | - | - | - |
| | Human Respiratory syncytial virus A | ATCC VR41 | - | - | - | - |
| | Influenza A H1N1 | KBPV VR33 | - | - | - | - |
| | Influenza A H1N1 | ATCC VR219 | - | - | - | - |
| | Influenza A H1N1 | ATCC VR897 | - | - | - | - |
| | Influenza A H1N1 | ATCC VR1683 | - | - | - | - |
| | Influenza A H3N2 | ATCC VR547 | - | - | - | - |
| | Influenza A H3N2 | ATCC VR822 | - | - | - | - |
| | Influenza A H3N2 | ATCC VR810 | - | - | - | - |
| | Influenza A H3N2 | ATCC VR544 | - | - | - | - |

RB4 mRT-PCR assays were performed using 69 strains from public centers corresponding to target species. The clinical isolates were identified by MALDI-TOF at the Department of Laboratory medicine in SGMF

Abbreviations: ATCC, American Type Culture Collection; KCTC, Korean Collection for Type Cultures; KCCM, Korean Culture Center of Microorganisms; KBPV, Korea Bank for Pathogenic Viruses; SGMF, Seegene Medical Foundation.

## Determination of analytical sensitivity

Analytical sensitivity and LOD were estimated with 50 replicates of positive control bacteria at six different concentrations, from 1 to $10^3$ copies/reaction (Table 3). The levels of 95% LOD were obtained using probit analysis: 143.88 copies/reactions for *K. pneumoniae*, 212.32 copies/reactions for *P. aeruginosa*, 166.72 copies/reactions for *S. aureus*, and 73.11 copies/reactions for *M. catarrhalis*. The RB4 mRT-PCR assay exhibited 100% reproducibility for all target bacteria as low as $5 \times 10^2$ copies/reaction, except for *M. catarrhalis*, where the assays were approximately five times more sensitive. The LOD was approximately $10–10^2$ copies/reaction, depending on the targets.

## Determination of analytical repeatability

Repeatability was measured by comparing bacterial loads of 100 replicates using the RB4 mRT-PCR assay: 20 positive controls at four different concentrations, high ($10^3$ copies/reaction), medium ($5 \times 10^2$ copies/reaction), low ($10^2$ copies/reaction), and very low ($5 \times 10^1$ copies/reaction), and 20 negative controls (Table 4). The number of positive tests for four target pathogens, out of total 80 replicates, were 80 (100%) at high, 80 (100%) at medium, 61 (76.3%) at low, and 25 (31.3%) at very low concentrations. The best repeatability was obtained from *M. catarrhalis* detection; 77 replicates were quantified as positive in all tested concentrations, resulting in a 96.3% repeatability. The repeatability values for *K. pneumoniae*, *P. aeruginosa*,

**Table 3. Evaluation of detection limit of target pathogens.**

| Pathogen | DNA Conc. (Copies/rxn) | Reactions | Positive | Positive rate (%) | LOD 95% level (Copies/rxn) |
|---|---|---|---|---|---|
| K. pneumoniae | $1 \times 10^3$ | 50 | 50 | 100 | 143.88 |
| | $5 \times 10^2$ | 50 | 50 | 100 | |
| | $1 \times 10^2$ | 50 | 45 | 90 | |
| | $5 \times 10^1$ | 50 | 16 | 32 | |
| | $1 \times 10^1$ | 50 | 0 | 0 | |
| | 1 | 50 | 0 | 0 | |
| P. aeruginosa | $1 \times 10^3$ | 50 | 50 | 100 | 212.32 |
| | $5 \times 10^2$ | 50 | 50 | 100 | |
| | $1 \times 10^2$ | 50 | 25 | 50 | |
| | $5 \times 10^1$ | 50 | 0 | 0 | |
| | $1 \times 10^1$ | 50 | 0 | 0 | |
| | 1 | 50 | 0 | 0 | |
| S. aureus | $1 \times 10^3$ | 50 | 50 | 100 | 166.72 |
| | $5 \times 10^2$ | 50 | 50 | 100 | |
| | $1 \times 10^2$ | 50 | 34 | 68 | |
| | $5 \times 10^1$ | 50 | 3 | 6 | |
| | $1 \times 10^1$ | 50 | 0 | 0 | |
| | 1 | 50 | 0 | 0 | |
| M. catarrhalis | $1 \times 10^3$ | 50 | 50 | 100 | 73.11 |
| | $5 \times 10^2$ | 50 | 50 | 100 | |
| | $1 \times 10^2$ | 50 | 50 | 100 | |
| | $5 \times 10^1$ | 50 | 42 | 84 | |
| | $1 \times 10^1$ | 50 | 11 | 22 | |
| | 1 | 50 | 0 | 0 | |

RB4 mRT-PCR reactions performed using serially diluted positive controls. The LOD 95% data were calculated using the probit model.

Abbreviations: Conc., concentration; rxn, reaction; LOD, limit of detection.

and *S. aureus* were 81.3% (65/80), 62.5% (50/80), and 67.5% (54/80), respectively. Overall, the results indicated that the RB4 mRT-PCR assay accurately quantifies the target bacterial load from positive controls at a concentration as low as $5 \times 10^2$ copies/reaction.

## Performance evaluation in clinical specimens

In the clinical test of 210 specimens, 73 and 137 sputum specimens obtained from pneumonia patients were negative and positive in reference assays, respectively. The sensitivity and NPVs of the RB4 mRT-PCR assay, compared to reference assays, were 100% for all target pathogens, and the specificity was 92.36% to *K. pneumoniae*, 85.71% to *P. aeruginosa*, 96.13% to *S. aureus*, and 98.96% to *M. catarrhalis*. PPVs were 85.71% for *K. pneumoniae*, 77.78% for *P. aeruginosa*, 90.16% for *S. aureus*, and 90.00% for *M. catarrhalis* (Table 5). Together, the RB4 mRT-PCR assay, compared to other standard methods, showed a better performance in clinical samples.

## Discussion

Clinical specimens from pneumonia patients frequently contain coinfections, and mRT-PCR has an advantage to simultaneously diagnose pathogens in a rapid and accurate manner [19, 20]. In this study, we developed and evaluated the RB4 mRT-PCR assay for simultaneous detection of four different bacterial pathogens causing pneumonia, including *K. pneumonia*, *P.*

**Table 4. Evaluation of detection repeatability of pathogens using mRT-PCR.**

| Pathogen | Conc. | Reactions | Positive | Mean ± SD | CV (%) |
|---|---|---|---|---|---|
| *K. pneumoniae* | High | 20 | 20 | 32.78 ± 0.22 | 0.67 |
| | Medium | 20 | 20 | 33.77 ± 0.32 | 0.95 |
| | Low | 20 | 18 | 36.35 ± 0.69 | 0.66 |
| | Very low | 20 | 7 | 37.48 ± 0.30 | 0.29 |
| | negative | 20 | 0 | - | - |
| *P. aeruginosa* | High | 20 | 20 | 34.00 ± 0.19 | 0.56 |
| | Medium | 20 | 20 | 35.48 ± 0.48 | 1.36 |
| | Low | 20 | 10 | 37.68 ± 0.21 | 0.57 |
| | Very low | 20 | 0 | - | - |
| | negative | 20 | 0 | - | - |
| *S. aureus* | High | 20 | 20 | 34.05 ± 0.20 | 0.59 |
| | Medium | 20 | 20 | 35.01 ± 0.21 | 0.60 |
| | Low | 20 | 13 | 37.37 ± 0.33 | 0.87 |
| | Very low | 20 | 1 | 37.56 | - |
| | negative | 20 | 0 | - | - |
| *M. catarrhalis* | High | 20 | 20 | 32.04 ± 0.16 | 0.50 |
| | Medium | 20 | 20 | 32.99 ± 0.23 | 0.70 |
| | Low | 20 | 20 | 35.22 ± 0.43 | 1.22 |
| | Very low | 20 | 17 | 36.74 ± 0.63 | 1.71 |
| | negative | 20 | 0 | - | - |

RB4 mRT-PCR reactions performed using positive controls in the amount of $10^3$, $5 \times 10^2$, $10^2$, and $5 \times 10^1$ copies/reaction corresponding to high, medium, low, and very low concentrations, respectively.

Abbreviations: Conc., concentration; SD, standard deviation; CV, coefficient of variation

**Table 5. Comparison of two platforms for the clinical qualification of target pathogens.**

| Pathogen | RB4 mRT-PCR | Reference assays | | | Kappa Value | *p* | Sen | Spe | PPV | NPV | DA |
|---|---|---|---|---|---|---|---|---|---|---|---|
| | | Positive | Negative | Total | | | | | | | |
| *K. pneumoniae* | Positive | 66 | 11 | 77 | 0.884 | <0.001 | 100 | 92.36 | 85.71 | 100 | 94.76 |
| | Negative | 0 | 133 | 133 | | | | | | | |
| | Total | 66 | 144 | 210 | | | | | | | |
| *P. aeruginosa* | Positive | 70 | 20 | 90 | 0.800 | <0.001 | 100 | 85.71 | 77.78 | 100 | 90.48 |
| | Negative | 0 | 120 | 120 | | | | | | | |
| | Total | 70 | 140 | 210 | | | | | | | |
| *S. aureus* | Positive | 55 | 6 | 61 | 0.929 | <0.001 | 100 | 96.13 | 90.16 | 100 | 97.14 |
| | Negative | 0 | 149 | 149 | | | | | | | |
| | Total | 55 | 155 | 210 | | | | | | | |
| *M. catarrhalis* | Positive | 18 | 2 | 20 | 0.942 | <0.001 | 100 | 98.96 | 90.00 | 100 | 99.05 |
| | Negative | 0 | 190 | 190 | | | | | | | |
| | Total | 18 | 192 | 210 | | | | | | | |

Analysis of Kappa value, *p*-value, sensitivity, specificity, positive predictive value, negative predictive value, diagnostic accuracy assessed using the RB4 mRT-PCR assay and reference assay in different target pathogens.

Abbreviations: Sen, sensitivity; Spe, specificity; PPV, positive predictive value; NPV, negative predictive value; DA, diagnostic accuracy.

*aeruginosa*, *S. aureus*, and *M. catarrhalis*. The performance of the RB4 mRT-PCR assay had analytical specificity for the four pathogens. The consistency of reference assays was >0.80 (kappa value, P < 0.001), and the clinical performance presented 100% reliability for all target pathogens (Table 5).

Sputum specimen retrieval is better accepted by patients for the diagnosis of LRTIs because sputum can be obtained easily and noninvasively [21]. Therefore, the RB4 mRT-PCR assay has been validated for use on sputum specimens rather than nasopharyngeal swabs or aspirates. Moreover, sputum specimens contain housekeeping genes that are stably preserved in human cells and can serve as internal controls [22]. The housekeeping gene *HBB* was used to ensure proper sampling and adequate target amplification for internal quality [23]. Although microscopy can determine sputum quality based on the number of polymorphonuclear leukocytes (≥10) and squamous epithelial cells (<25) [24], the use of *HBB* as an internal control correctly determines the quality of the specimen, procedure of nucleic acid extraction, and presence of PCR inhibition [25, 26].

The advantage of the RB4 mRT-PCR assay is its effectiveness on both reference materials and clinical specimens and its correct determination of all positive and negative strains (Table 2). Moreover, 10 (six single and four mixed infections) of the 73 (13.7%) negative specimens and 118 of the 137 (86.1%) positive specimens were matched, whereas RB4 mRT-PCR assays detected an additional 19 bacterial infections from reference-positive samples: 15 (78.9%) single infections and four (21.1%) mixed infections (S1 Table). The RB4 mRT-PCR assay procedure includes automated sample processing and internal specimen quality check, and it can report results within 4 h of specimen receipt on a standard mRT-PCR platform.

There are several noteworthy points in this study. (1) This assay lacks quantitative molecular levels and correlation between Ct values and colony-forming units of bacterial pathogens for LRTI diagnosis; however, it showed stable Ct values with 0.29–1.71% CV (Table 4) [9, 27]. These results indicated that the RB4 mRT-PCR platform is more sensitive than reference assays, even at low copy numbers (Table 3). (2) The assay results were validated with sputum specimens and not with other LRTI specimens, such as endotracheal aspirate and bronchoalveolar lavage [28]. There is a need to improve the efficiency of the mRT-PCR assay and develop test kits compatible with a broad range of LRTI specimens.

The RB4 mRT-PCR assay can potentially serve as an improved decision-making tool during LRTI treatment. Faster and more accurate diagnosis of pathogens would promote the use of narrow- over broad-spectrum antibiotics and substantially reduce the inappropriate use of antibiotics.

## Supporting information

**S1 Table. Comparison of discordant samples between reference assays and the RB4 mRT-PCR assays.** The RB4 mRT-PCR assays detected an additional 19 bacterial infections from reference-positive assays.
(PDF)

## Acknowledgments

We would like to thank Editage (www.editage.co.kr) for English language editing.

## Author Contributions

**Conceptualization:** Min Young Park, Nackmoon Sung, Yong-Jin Yang.

**Data curation:** Ho Jae Lim, Eun-Rim Kang, Min Young Park, Bo Kyung Kim.

**Formal analysis:** Ho Jae Lim, Eun-Rim Kang, Min Jin Kim, Sunkyung Jung, Kyoung Ho Roh, Jae-Hyun Yang, Min-Woo Lee, Sun-Hwa Lee, Yong-Jin Yang.

**Investigation:** Ho Jae Lim, Eun-Rim Kang, Min-Woo Lee, Yong-Jin Yang.

**Methodology:** Ho Jae Lim, Eun-Rim Kang, Nackmoon Sung, Yong-Jin Yang.

**Supervision:** Min Young Park, Yong-Jin Yang.

**Validation:** Ho Jae Lim, Min Young Park, Bo Kyung Kim, Jae-Hyun Yang, Min-Woo Lee, Yong-Jin Yang.

**Writing – original draft:** Ho Jae Lim, Jae-Hyun Yang, Yong-Jin Yang.

**Writing – review & editing:** Min Young Park, Min Jin Kim, Sunkyung Jung, Kyoung Ho Roh, Nackmoon Sung, Jae-Hyun Yang, Min-Woo Lee, Sun-Hwa Lee, Yong-Jin Yang.

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
