## [Decision Letter · Decision Letter 0]

6 May 2021

PONE-D-21-10845

Development of a multiplex real-time PCR assay for the simultaneous detection of four bacterial pathogens causing pneumonia

PLOS ONE

Dear Dr. Yang,

Thank you for submitting your manuscript to PLOS ONE. After careful consideration, we feel that it has merit but does not fully meet PLOS ONE’s publication criteria as it currently stands. Therefore, we invite you to submit a revised version of the manuscript that addresses the points raised during the review process.

The reviewers have no significant comments on this work. Authors should look at the reviewers' comments and respond to these comments.

We look forward to receiving your revised manuscript.

Kind regards,

Ruslan Kalendar, PhD

Academic Editor

PLOS ONE

Journal Requirements:

3. Thank you for including your ethics statement: "This study was approved by the institutional review board (SMF-IRB-2021-001) of the SGMF."

a) Please provide additional details regarding participant consent. In the ethics statement in the Methods and online submission information, please ensure that you have specified (i) whether consent was informed and (ii) what type you obtained (for instance, written or verbal, and if verbal, how it was documented and witnessed).

If your study included minors, state whether you obtained consent from parents or guardians.

If the need for consent was waived by the ethics committee, please include this information.

Reviewers' comments:

Reviewer's Responses to Questions

**Comments to the Author**

1. Is the manuscript technically sound, and do the data support the conclusions?

Reviewer #1: Yes

Reviewer #2: Yes

2. Has the statistical analysis been performed appropriately and rigorously? 

Reviewer #1: Yes

Reviewer #2: Yes

3. Have the authors made all data underlying the findings in their manuscript fully available?

Reviewer #1: Yes

Reviewer #2: Yes

4. Is the manuscript presented in an intelligible fashion and written in standard English?

Reviewer #1: Yes

Reviewer #2: Yes

5. Review Comments to the Author

Reviewer #1: 

The manuscript was well designed and has technical rigor. The study question was answered. The proposed technique may help the diagnosis of pneumonia and will prevent the indiscriminate use of antibiotics.

Reviewer #2: 

The new mPCRs are promising tools to help clinicians to make clinical decisions in antibiotic treatments. The authors presented a promising array which could be useful in a clinical setting. It would be extremely useful information (and significantly strengthens the utility of the assay) to know if the assay is able to differentiate antibiotic resistant strain of the different types of bacteria.

1) Any analysis for hairpins etc for the PCR primers?

2) Have you investigated your assay against antibiotic resistant strains?

3) Have you investigated the antibiotic susceptibility data against the results from the assay?

6. PLOS authors have the option to publish the peer review history of their article (what does this mean?). If published, this will include your full peer review and any attached files.

Reviewer #1: No

Reviewer #2: No

---

## [Author Response · Author response to Decision Letter 0]

31 May 2021

Reviewers' comments:

Reviewer's Responses to Questions

Comments to the Author

1. Is the manuscript technically sound, and do the data support the conclusions?

Reviewer #1: Yes

Reviewer #2: Yes

2. Has the statistical analysis been performed appropriately and rigorously? 

Reviewer #1: Yes

Reviewer #2: Yes

3. Have the authors made all data underlying the findings in their manuscript fully available?

Reviewer #1: Yes

Reviewer #2: Yes

4. Is the manuscript presented in an intelligible fashion and written in standard English?

Reviewer #1: Yes

Reviewer #2: Yes

5. Review Comments to the Author

Reviewer #1: 

The manuscript was well designed and has technical rigor. The study question was answered. The proposed technique may help the diagnosis of pneumonia and will prevent the indiscriminate use of antibiotics.

Reviewer #2: 

The new mPCRs are promising tools to help clinicians to make clinical decisions in antibiotic treatments. The authors presented a promising array which could be useful in a clinical setting. It would be extremely useful information (and significantly strengthens the utility of the assay) to know if the assay is able to differentiate antibiotic resistant strain of the different types of bacteria.

1) Any analysis for hairpins etc for the PCR primers?

A: We checked the secondary structures of oligomers (primers and probes) using the GeneRunner software. We tested all 15 oligomers used in this study, and five showed no secondary structures. However, the predicted secondary structures did not significantly affect our PCR-based methods using dual-priming oligonucleotides (DPO) because the 30-mer oligo alone, unlike the 50-mer oligo (#1, 2), was not stable enough to form secondary structures. The GC content (<65 %) and the avoidance of repetitive sequences met the oligo design criteria (#3).

- References –

1. Chun JY, Kim KJ, Hwang IT, Kim YJ, Lee DH, Lee IK, et al. Dual priming oligonucleotide system for the multiplex detection of respiratory viruses and SNP genotyping of CYP2C19 gene. Nucleic Acids Res. 2007;35(6):e40. Epub 2007/02/09. doi: 10.1093/nar/gkm051. PubMed PMID: 17287288; PubMed Central PMCID: PMCPMC1874606.

2. Fredman D, Jobs M, Strömqvist L, Brookes AJ. DFold: PCR design that minimizes secondary structure and optimizes downstream genotyping applications. Hum Mutat. 2004;24(1):1-8. doi: 10.1002/humu.20066. PMID: 15221783.

3. Riet J, Ramos LRV, Lewis RV, Marins LF. Improving the PCR protocol to amplify a repetitive DNA sequence. Genet Mol Res. 2017;16(3). doi:10.4238/gmr16039796.

2) Have you investigated your assay against antibiotic resistant strains?

A: Yes, RB4 mRT-PCR assays worked for antibiotic-resistant pathogens. Fifty-eight samples used in this study were tested for antibiotic resistance as a part of the minimal inhibitory concentration (MIC) test procedure. 

<Data table>

Pathogen Total samples (N) MIC test (%) Antibiotic-resistant (N)

K. pneumoniae 66 20 (30.3%) 20

P. aeruginosa 70 29 (41.4%) 27

S. aureus 55 10 (18.2%) 10

M. catarrhalis 18 0 (0%) 0

 Data are representative of the total samples, MIC test, and antibiotic resistance for target pathogens in 58 samples.

3) Have you investigated the antibiotic susceptibility data against the results from the assay?

A: Yes. K. pneumoniae, P. aeruginosa, and S. aureus were analyzed by MIC tests for resistance to 37 antibiotics and classified into three categories: antibiotic-resistant, antibiotic-intermediate, or antibiotic-susceptible. We analyzed the association of antibiotic resistance with extended-spectrum beta-lactamase (ESBL) gene types in K. pneumoniae strains. We screened 17, 17, and 16 antibiotics for K. pneumoniae, P. aeruginosa, and S. aureus, respectively.

<Data table>

Antibiotic K. pneumoniae P. aeruginosa S. aureus

Amikacin S S N/A

Amoxicillin/Clavulanic acid R N/A N/A

Ampicillin R N/A N/A

Ampicillin/Sulbactam N/A R N/A

Aztreonam R S N/A

Cefazolin R N/A N/A

Cefepime R S N/A

Cefotaxime R R N/A

Cefoxitin S N/A N/A

Ceftazidime R S N/A

Ciprofloxacin R S R

Clindamycin N/A N/A R

Colistin N/A S N/A

Ertapenem S N/A N/A

Erythromycin N/A N/A R

Esbl + N/A N/A

Fusidic acid N/A N/A S

Gentamicin S S S

Imipenem S S N/A

Levofloxacin N/A N/A N/A

Linezolid N/A N/A S

Meropenem N/A S N/A

Mikacin N/A N/A N/A

Minocycline N/A R N/A

Oxacillin N/A N/A R

Penicillin G N/A N/A R

Piperacillin N/A I N/A

Piperacillin/Tazobactam R I N/A

Quinupristin/Dalfopristin N/A N/A S

RFP (Rifampicin) N/A N/A S

Teicoplanin N/A N/A S

Telithromycin N/A N/A R

Tetracycline N/A N/A S

Ticarcillin/Clavulanic acid N/A S N/A

Tigecycline S R S

Tobramycin N/A N/A N/A

Trimethoprim/Sulfamethoxazole S R S

Vancomycin N/A N/A S

Data are representative of antibiotics for target pathogens in 58 samples. Abbreviations: R; resistant; I, intermediate; S, susceptible; N/A; not available.

We hope you are satisfied with our answer.

Thank you very much. 

6. PLOS authors have the option to publish the peer review history of their article (what does this mean?). If published, this will include your full peer review and any attached files.

Do you want your identity to be public for this peer review? For information about this choice, including consent withdrawal, please see our Privacy Policy.

Reviewer #1: No

Reviewer #2: No

---

## [Editor Report · Decision Letter 1]

4 Jun 2021

Development of a multiplex real-time PCR assay for the simultaneous detection of four bacterial pathogens causing pneumonia

PONE-D-21-10845R1

Dear Dr. Yang,

We’re pleased to inform you that your manuscript has been judged scientifically suitable for publication and will be formally accepted for publication once it meets all outstanding technical requirements.

Kind regards,

Ruslan Kalendar, PhD

Academic Editor

PLOS ONE

---

## [Editor Report · Acceptance letter]

9 Jun 2021

PONE-D-21-10845R1 

Development of a multiplex real-time PCR assay for the simultaneous detection of four bacterial pathogens causing pneumonia 

Dear Dr. Yang:

I'm pleased to inform you that your manuscript has been deemed suitable for publication in PLOS ONE. Congratulations! Your manuscript is now with our production department. 

Kind regards, 

on behalf of

Prof. Ruslan Kalendar 

Academic Editor

PLOS ONE